# Expansion of *Kuravirus*-like Phage Sequences within the Past Decade, including *Escherichia* Phage YF01 from Japan, Prompt the Creation of Three New Genera

**DOI:** 10.3390/v15020506

**Published:** 2023-02-11

**Authors:** Steven Batinovic, Yugo Fujii, Tadashi Nittami

**Affiliations:** Division of Materials Science and Chemical Engineering, Yokohama National University, Yokohama 240-8501, Kanagawa, Japan

**Keywords:** *Kuravirus*, *Escherichia* phage, nanopore phage, phage genomics, ICTV

## Abstract

Bacteriophages, viruses that infect bacteria, are currently receiving significant attention amid an ever-growing global antibiotic resistance crisis. In tandem, a surge in the availability and affordability of next-generation and third-generation sequencing technologies has driven the deposition of a wealth of phage sequence data. Here, we have isolated a novel *Escherichia* phage, YF01, from a municipal wastewater treatment plant in Yokohama, Japan. We demonstrate that the YF01 phage shares a high similarity to a collection of thirty-five *Escherichia* and *Shigella* phages found in public databases, six of which have been previously classified into the *Kuravirus* genus by the International Committee on Taxonomy of Viruses (ICTV). Using modern phylogenetic approaches, we demonstrate that an expansion and reshaping of the current six-membered *Kuravirus* genus is required to accommodate all thirty-six member phages. Ultimately, we propose the creation of three additional genera, *Vellorevirus*, *Jinjuvirus*, and *Yesanvirus*, which will allow a more organized approach to the addition of future *Kuravirus*-like phages.

## 1. Introduction

*Escherichia coli* is a versatile and genetically diverse species commonly inhabiting the intestinal tracts of humans and animals as both a commensal and pathogenic microbe [1]. Pathogenic *E. coli* infecting humans are routinely released into the environment via sewage systems, which lead to wastewater treatment plants (WWTPs) [2]. Given the rising global public health concerns surrounding antibiotic resistance in pathogenic microbes, such as *E. coli*, alternative methods to control these pathogens are highly sought after. Bacteriophages (or phages) have received high attention as a potential therapeutic alternative to antibiotics [3,4,5]. 

Phages are viruses that multiply by infecting bacteria, often very specifically at the species or strain level. They play a key role both in microbial ecology and microbial evolution, having the ability to alter the population dynamics within microbial communities and modify bacterial genomes through horizontal gene transfer [5,6,7]. In biological WWTPs, phage concentrations are estimated to be approximately 10^8^–10^9^ particles per milliliter [8,9,10,11], which is higher than many other ecosystems studied to date [6,10]. Studies have shown that some of phage agents can control the growth of *E. coli* in biological WWTPs [12]. The use of multiple phages (phage cocktail) can more effectively control phage resistance and the recovery of phage-resistant bacteria after treatment [13]. 

The advent of next-generation sequencing technologies, such as those offered by Illumina, has led to a preponderance of genomic sequencing. Newer “third-generation sequencing” technologies, such as those offered by Oxford Nanopore, facilitate the acquisition of long-sequence reads (up to Mbs), allowing for the generation of complete bacterial and other higher organism genomes. While commonly used in combination with highly accurate short-reads, more recent revisions of the technology, known as Q20, include qualitative improvements that facilitate the generation of near-perfect bacterial assemblies, and perfect viral assemblies, without the need for short-read polishing [14,15].

As of December 2022, there are currently ~1600 *Escherichia* phage assemblies listed on NCBI Genbank. The International Committee on Taxonomy of Viruses (ICTV) currently recognizes *Escherichia* phages belonging to at least 11 different families and 100 different genera [16]. Given the wealth of sequencing currently available, the genomic diversity of phages and the way we assign their taxonomy is constantly shifting and evolving. For example, the *Phieco32-like virus* genus was created in 2009, in response to the isolation and characterization of the *Escherichia* phage phiEco32, a novel 77 kb phage with a rare C3 podovirus morphotype [17]. The genus was subsequently renamed twice within the next few years (to *Phieco32likevirus* and then *Phieco32virus*), while also adding five new phage members to its ranks (*Escherichia* viruses ECBP2, NJ01, Septima11, SU10, and kv1721) [18,19,20]. The genus was again renamed to its current name of *Kuravirus* in 2019.

In this study, we isolated a novel *E. coli* phage, YF01, from a WWTP in Yokohama, Japan. The YF01 phage was sequenced and assembled using long-read Q20 Oxford Nanopore technology and shown to be related to phages from the ICTV-classified *Kuravirus* genus. Modern reticulate network analyses revealed that YF01 and the current six ICTV-classified *Kuravirus* phages shared high similarity with 29 other phages present in public databases. Using a combination of approaches, we demonstrate that these 36 phages represent four different genera under current ICTV guidelines. We further analyze the genomic features of the YF01 phage in comparison to other *Kuravirus*-like group phages and elucidate the core proteome of the group to demonstrate that the large terminase subunit and portal protein, among three other proteins, show high conservation and highly correlated phylogenetic reconstruction to whole genome-based methods. Ultimately, we recommend the expansion and reshaping of the *Kuravirus* genus to four separate genera to accommodate a total of 36 phages isolated worldwide.

## 2. Materials and Methods

### 2.1. Bacterial Strains and Media

*Escherichia coli* strain K12 (JCM20135), isolated from feces from a diphtheria convalescent sample, was used in this study. Bacterial cultures were grown in Nutrient Broth with NaCl (5 g L^−1^ peptone, 3 g L^−1^ beef extract and 5 g L^−1^ NaCl, pH 7; NB) and NB with agar (with added 12 g L^−1^ agar) at 25 °C. Cultures were grown under aerobic conditions.

### 2.2. Isolation and Purification of Phages

Activated sludge mixed liquor was collected from a municipal WWTP in Yokohama in November 2021 where urban wastewater was received. The activated sludge sample was transferred to a 15 mL tube and centrifuged at 2000× *g* for 5 min. The supernatant was filtered through a cellulose acetate (CA) filter with a 0.20 μm pore size (Advantec Toyo, Tokyo, Japan). A total of 1 mL of filtrate was added to NB and 100 µL of log phase *E. coli* K12. The suspension was incubated overnight at 25 °C. The overnight culture was centrifuged at 2000× *g* for 10 min and the supernatant filtered through a CA 0.20 μm pore filter (Advantec Toyo, Tokyo, Japan). *E. coli* colonies from a growth plate were taken with sterile cotton swab and spread uniformly on NB agar medium. A total of 40 μL of filtrate was added plate dropwise, air-dried, and incubated at 25 °C overnight. A visible plaque was removed with the wide end of a tip and resuspended in SM buffer (200 mM NaCl, 10 mM MgSO_4_, 50 mM Tris-HCL, pH 7.5), a process repeated twice to ensure a pure phage isolate. 

### 2.3. Transmission Electron Microscopy

TEM was performed as previously described [21]. Briefly, copper grids coated with carbon and formvar were first subjected to a glow discard treatment for 60 s. A total of 20 µL the YF01 phage (~1 × 10^11^ PFU mL^−1^) was incubated on a grid for 10 min. Grids were washed twice in MilliQ water and stained with 2% (*w*/*v*) uranyl acetate prior to a final wash in MilliQ water. Grids were dried prior to examination under a JEOL JEM02010HC electron microscope. 

### 2.4. Verification of pH and Temperature Stability

To test phage stability, the pH of NB was adjusted to 3, 5, 7, 9, and 11 using NaOH and HCl. The pH-adjusted NB and YF01 phages (~1 × 10^11^ PFU mL^−1^) were mixed in a 9:1 ratio and incubated at 25 °C for 1 h. Phage PFU mL^−1^ was then determined by spot test (20 µL) using a dilution series down to 10^−8^ and counting the dilution that resulted in 10–100 plaques. For temperature testing, the NB and YF01 phages (~1 × 10^11^ PFU mL^−1^) were mixed in a 9:1 ratio and incubated for 1 h at different temperatures (4, 25, 37, 50, 60, 70, and 80 °C). Phage PFU mL^−1^ was determined as above. Each experiment was repeated two times.

### 2.5. Phage DNA Isolation

Genomic DNA from 1 mL phage filtrate (>1 × 10^10^ PFU mL^−1^) was extracted using a zinc chloride phenol:chloroform extraction method [22]. Briefly, the phage filtrate was first treated with DNase I, Rnase A, and MgCl_2_ to remove host contaminants before the precipitation of phage virions by the addition of 40 mM ZnCl_2_. Virions were resuspended in phage extraction buffer (400 mM NaCl, 20 mM EDTA, 0.5% (*w*/*v*) SDS and 50 µg mL^−1^ proteinase K) and incubated for 1 h at 55 °C. An equal volume of phenol:chloroform:isoamyl alcohol (25:24:1) was added and the top layer was removed and DNA precipitated with isopropanol. The pellet was washed in 70% (*v*/*v*) ethanol and resuspended in 10 mM Tris-HCl (pH 8.5). DNA concentration and purity were measured using a Quantus Fluorometer (Promega, Tokyo, Japan) and NanoDrop One (Thermo Scientific, Tokyo, Japan), respectively. DNA integrity was assessed via agarose gel electrophoresis.

### 2.6. Phage DNA Sequencing and Assembly

DNA libraries were prepared using 1 ug of DNA using the Ligation Sequencing Kit 14 (SQK-LSK114), loaded on a R10.4.1 flow cell (FLO-MIN114) and sequenced using a MinION Mk1B (Oxford Nanopore, Tokyo, Japan). Simplex and duplex basecalling was performed using Guppy v6.4.2 in super accurate (SUP) mode. Read filtering was performed with Filtlong v0.2.1 (https://github.com/rrwick/Filtlong). Filtered mean read size was 17.8 kb with a mean read quality of Q20.3, as assessed by NanoPlot v1.40.0 [23]. Reads were assembled with Flye v2.9.1 [24] in high-quality mode to generate a complete phage assembly (704X fold coverage). Post-assembly polishing with Medaka v1.7.2 (https://github.com/nanoporetech/medaka) made no corrections to the assembly. Residual Nanopore adapter sequences were truncated from the termini of the final assembly by screening against Y-adapter top (5′-GGCGTCTGCTTGGGTGTTTAACCTTTTTTTTTTAATGTACTTCGTTCAGTTACGTATTGCT-3′) and Y-adapter bottom (5′- GCAATACGTAACTGAACGAAGT -3′) sequences.

### 2.7. Phylogenetic Analysis

For the vConTACT2 reticulate network analysis, the YF01 phage genome was first annotated with Prokka v1.14.6 [25] and the phage protein sequences were combined with protein sequences from all phage genomes present on Genbank in November 2022 (19,164 genomes) [26]. Gene2Genome was used to assign and map protein coding sequences prior to the use of vConTACT2 v0.9.19 using default settings [27]. The network was visualised with Cytoscape v3.9.1 [28] using the default layout for the entire network and then an edge-weighted, spring-embedded model in the zoom view. *Kuravirus*-like phage group matrix nucleotide similarity analyses were performed using VIRIDIC v1.0 [29] and visualised with R v4.0.3 and R Studio v2022.07.1 using the pheatmap package v.1.0.12. For VICTOR analysis, *Kuravirus*-like group protein sequences were generated using Prokka as above and analysed using the VICTOR3 pipeline [30]. Briefly, pairwise comparisons of the nucleotide sequences were conducted using the Genome-BLAST Distance Phylogeny (GBDP) method [31] under settings recommended for prokaryotic viruses [30]. The resulting intergenomic distances were used to infer a minimum evolution tree with branch support via FASTME including SPR postprocessing [32]. Branch support was inferred from 100 pseudo-bootstrap replicates each. Taxon boundaries at the species, genus, and family level were estimated through the OPTSIL program [33], the recommended clustering thresholds [30], and an F value (fraction of links required for cluster fusion) of 0.5 [34]. Phylogenetic trees inferred using the D0 formula (optimised for nucleotides) were displayed using iTOL v5 [35]. The phylogenetic reconstruction of group core proteins (terminase large subunit, portal protein, MCP, DNA helicase/primase and hypothetical protein were performed with Phylogeny.fr [36] using MUSCLE v3.8.31 alignment [37] and PhyML v3.1 using the maximum likelihood method for reconstruction [38].

### 2.8. Genome Annotation

For the genome termini analysis, raw >Q30 Nanopore sequencing reads above 78,000 bp were filtered with Filtlong v0.2.1. Reads were then manually inspected for the presence of direct terminal repeats (DTRs). DTR sequences in other *Kuravirus*-like group phages were identified using the “Annotate from…” function, as performed previously [22]. To annotate the YF01 phage genome, the assembly was imported into Geneious Prime v2022.2.1 and genes were predicted with Glimmer3 [39] and manually inspected for the presence of ribosome binding sites (RBS). ORFs were annotated using a combination of the NCBI Conserved Domain Database (CDD) [40], a profile hidden Markov model (HMM) similarity using Hhpred [41], and the Virfam webserver [42]. tRNAs were identified using tRNAscan-SE [43] and Aragorn v1.2.41 [44]. Figures were generated using CLC Genomics WorkBench v9.5.5 and Clinker v0.0.12 [45]. Amino acid similarity calculations were performed using Clustal Omega v1.2.3 [46] in Geneious Prime v2022.2.1 with the Blosum 62 similarity matrix. Average amino acid similarity values presented in the text refer to the average value of all similarity values in the matrix. Raw data and calculations are available in the Appendix A.

### 2.9. Codon Usage Analysis

Coding sequences (nucleotide) for YF01, phiEco32, ES17, KBNP1711, and ECBP2 phages were determined using Prokka and directly accessed from Genbank for *E. coli* K12 MG1655 (NC_000913). Codon usage was determined using the statistics function in Geneious Prime v2022.2.1. Codon usage that differed by ≥2-fold compared to the host were plotted using Prism v7.0.

### 2.10. Pangenome Analysis

The core analysis of the *Kuravirus*-like group phages was performed using Roary [22]. Briefly, phage genomes were first annotated with Prokka. Roary v3.13.0 core gene alignment [47] was used with a reduced BLASTp identity threshold of 30% [48] and the representative core proteome (YF01 phage) was subject to bioinformatic analyses (as described in section above). Charts were generated with Prism v7.0.

## 3. Results and Discussion

### 3.1. Isolation and Characteristics of Escherichia Phage YF01

*Escherichia* phage YF01 was isolated from activated sludge collected from a municipal WWTP in Yokohama, Japan. The phage formed small, transparent, circular plaques approximately 1 mm in diameter (Figure 1a). Phage virions negatively stained with uranyl acetate and imaged under the transmission electron microscope displayed the unusual C3 podovirus morphotype with an elongated capsid and short tail (Figure 1b). We next tested the stability of the YF01 phage by incubating it at varying temperature and pH conditions. The YF01 phage was stable at temperatures from 4–50 °C, with a decline in phage activity (as measured by PFU mL^−1^) noted from 60 °C and no observable phage activity above 70 °C (Figure 1c). The YF01 phage was relatively stable in the pH range of 3–11 with only mild reduction (~2-fold) in phage activity noted after 1 h of storage at pH 3 (Figure 1d). The pH of activated sludge systems are typically rather neutral (~6.5–8 pH), and even with the activated sludge at the plant in Yokohama recording at the lower end of this range (pH 6.4), the YF01 phage is well suited to persisting in wastewater environments. 

### 3.2. Genome Assembly and Phylogenetic Placement of the YF01 Phage

The YF01 phage was sequenced using long-read Q20 Oxford Nanopore technology, leading to the assembly of a 78,626 bp genome with 704-fold coverage (Table 1). The GC content of the YF01 phage, 42.1%, was markedly lower than that observed of its host *E. coli* (~50.8%), a common phenomenon among lytic phages [49].

To phylogenetically place the YF01 phage we first made use of vContact2, which is a network-based approach that measures the degree of protein content shared between viruses to compute viral clusters (VC) [27]. As vContact2 allows for the input of up to 1 million sequences, a global network phage analysis was performed by inputting the total protein content of all complete phage genome sequences present within the Genbank database as of November 2022 (19,164 phage sequences) in addition to the YF01 phage. In the resulting reticulate network (Figure 2), phages that were found to share statistically confident protein-level similarity with the YF01 phage (as indicated by edge-connection) were further extracted and visualized using an edge-weighted layout to observe their grouping more accurately (Figure 2; zoom panel). The YF01 phage (in blue) formed a VC with 34 other *Escherichia* (and 1 *Shigella*) phages (in yellow; Figure 2). Six of these phages, phiEco32 [17], ECBP2 [18], NJ01 [19], KBNP1711, SU10 [20], and 172-1, were previously classified by the ICTV as the *Kuravirus* genus. vContact2 further split the 36-membered VC into two sub-VCs (considered to be the equivalent to genera grouping [27]) of 24 and 12 phages, with YF01, phiEco32, KBNP1711, SU10, and 172-1 phages and 18 non-ICTV-classified phages placed in the former and the ECBP2 phage and 11 non-ICTV-classified phages placed in the latter. Phages infecting other bacterial genera within the *Gammaproteobacteria*, such as *Aeromonas* phage Lah_6, *Proteus* phage Privateer, *Salmonella* phage 7-11, and *Vibrio* phage Vp_R1, showed the highest protein-level similarity to the 36-membered VC (Figure 2; zoom panel).

We took these 36 grouped phages, including the YF01 phage and the six ICTV-classified *Kuravirus* phages, and performed matrix nucleotide identity analysis to understand the nucleotide-level similarity more precisely. Using nucleotide similarity thresholds of 95% and 70%, for species and genus demarcation, respectively, as recommended by the ICTV, VIRIDIC grouped these phages into a total of thirty-three species and four genera G1-G4 (Figure 3; Appendix A). In contrast to the genera placings suggested by vContact2 (sub-VCs), here we saw a further split of the first sub-VC (24 phages) into three separate genera of nineteen (G1), three (G2), and two (G3) phages. Four of the ICTV-classified *Kuravirus* phages (phiEco32, NJ01, SU10, and 172-1 phages) grouped into the largest genus (G1), while KBNP1711 and ECBP2 phages grouped into two separate genera, G3 and G4, respectively. The apparent discrepancy between our genera groupings and the *Kuravirus* classifications by ICTV occurred due to the nucleotide similarity between the KBNP1711 and ECBP2 phages and the other ICTV-classified phages (such as the phiEco32 phage) falling below the currently accepted threshold of 70% (i.e., ~60.9% nucleotide similarity ECBP2 vs. phiEco32).

We then performed a phylogenetic reconstruction of the clustered phage genomes using VICTOR, which is based on Genome BLAST Distance Phylogeny (GBDP), using whole phage genome sequence inputs [30]. Here we input the complete nucleotide sequence of the 36 clustered phages as well as the complete nucleotide sequences from 14 related phages infecting other *Aeromonas*, *Cronobacter*, *Escherichia*, *Proteus*, *Salmonella*, and *Vibrio* species. VICTOR phylogeny demonstrated that the 36 clustered phages formed a monophyletic group (Figure 4), with clear sub-clade formation representing the four genus groupings G1–G4, indicated by VIRIDIC (Figure 4, G1–4). OPTSIL taxon boundary prediction suggested, in contrast to our vContact2 and VIRIDIC analyses, that the 36 grouped phages belonged to a single genus. VICTOR was originally optimized against a large reference phage genome database recognized by the ICTV in 2017, and it is likely that the evolution of threshold criteria and the rapid expansion of ICTV-classified phages over the past 5 years has caused this apparent divergence. 

The current ICTV classification of the six *Kuravirus* phages occurred between 2009 (the creation of the genus with phiEco32 phage) and 2015–2016 (the addition of ECBP2, NJ01, KBNP1711, SU10, and 172-1 phages). Given the substantial increase in the amount of *Kuravirus*-like phage genomes deposited in public databases since then (30 new *Kuravirus*-like phages deposited in the past 9 years) and based upon the current criteria for the genome-based classification of viruses [48], we believe it would be sensible to revisit and reshape the *Kuravirus* genus into four separate genera based on the currently accepted similarity thresholds (95% and 70% for species and genus, respectively). We will recommend to the ICTV the identification of these four genera as *Kuravirus* (G1), *Vellorevirus* (G2), *Jinjuvirus* (G3), and *Yesanvirus* (G4), with the names of the new genera (G2–G4) representing either the geographical location of isolation or locality of the research group, of the founding phage member of each respective genus (myPSH1131, KBNP1711, and ECB2 phages for *Valleorevirus*, *Jinjuvirus*, and *Yesanvirus*, respectively). At the present time we will refer to this grouping of 36 phages as the *Kuravirus*-like phage group or simply “the group” (Figure 4). 

### 3.3. Genomic Organisation of the YF01 Phage

The 78,626 bp YF01 phage genome encodes 121 CDS, 66% of which are hypothetical proteins, and a single tRNA Appendix A. The genome is characteristically organized into functional modules, with genes involved in virion morphogenesis and lysis module on the left (*gp3-gp22*) and genes involved in DNA replication and nucleotide metabolism on the right (*gp23-gp81*; Figure 5). 

#### 3.3.1. Genome Termini

*Kuravirus*-like group phages are known to exhibit short (~190 bp) direct terminal repeats (DTRs) indicative of a T7 phage-like replication strategy where the substrate for capsid packaging are large DNA concatemers. The presence of DTRs in the group was first determined in phiEco32 phage (193-bp DTRs) using a combination of restriction profiling, primer walking, and sub-cloning strategies [17]. In more modern approaches, DNA termini are often computed based off raw short-read sequence data via the analysis of starting positive coverage [63]. With this approach, the termini of only two other group phages have been determined, SU7 and Paul phages, with noted DTRs of 53 bp (cautioned by the authors) and 193 bp, respectively [56,61]. Given the YF01 phage was sequenced with nanopore long-read technology, we curated reads spanning the entire YF01 genome sequence (~78,600 bp; many >Q30 due to duplex strand basecalling) and analyzed the DNA termini. Except residual sequencing adapter sequence on the read termini, we noted the distinct presence of 193 bp direct repeats on most reads, consistent with our assembly, indicating that the YF01 phage, like phiEco32 and Paul phages, contains 193 bp DTRs. Using sequence similarity, we identified 192–193 bp DTRs in a total of 30 phages in the group, which share a highly conserved region between nucleotide positions ~130–170 bp Appendix A. 

#### 3.3.2. Virion Morphogenesis and Lysis

The virion morphogenesis and lysis module in the YF01 phage shares high homology with other group phages, comprising a total of 20 genes transcribed in the forward direction (Figure 6). The terminase small and large subunits (*gp3-4*), which form the molecular motor responsible for packaging of the nascent phage DNA, are the first two genes in this module. While *gp3* displayed no known domains via the NCBI conserved domains database (CDD), it did show similarity to the small terminase subunit in the *Pseudomonas* phage PaP3 using a more sensitive comparison of profile hidden Markov models (HMM) [41,64]. We did note that some members of the group, such as the 172-1 phage, have a gene further downstream of this module (equivalent to *gp25* in the YF01 phage) annotated as a terminase small subunit. This was likely due to the presence of a HNH endonuclease domain, which are often associated with phage DNA packaging [65]. However, given HNH endonuclease domains are numerous in phages (i.e., there are four in the YF01 phage) and the small and large terminase subunits are commonly co-located in the phage genome, we have instead assigned *gp3* as the putative small terminase subunit in the YF01 phage. 

The next genes in this module (*gp5-9*) encode proteins that assemble to form the rare C3 morphotype (an elongated phage capsid) seen in the YF01 phage (Figure 1b) and other group members, including the portal protein (*gp5*), scaffolding protein (*gp7*), major capsid protein (*gp8*), and adapter protein (*gp9*; Figure 6). The largest variance was observed in the major capsid protein (MCP) of the YF01 phage, compared to the representative members of the group (Figure 6). The MCP in the YF01 and other group phages [17] is transcribed in two forms; a shorter product corresponding to the natural start (ATG) and the stop (TAA) codons of *gp8* (nt. 7253–8314 in the YF01 phage, 353 a.a) and a larger product that occurs due to a ribosomal slippage event at a heptanucleotide slippery sequence (GGGAAAG, nt. 8293-8299 in the YF01 phage) near the natural stop codon of *gp8*, leading to an extended isoform using the -1 frame (nt. 7253-9930 in the YF01 phage, 892 a.a). This is a well-known phenomenon in the tail assembly chaperone in long-tailed phages (~3.5% slippage rate in the *Escherichia* phage Lambda [66]) but also occurs in the MCP of *Escherichia* phages T3 and T7 (~10% slippage rate) [67,68]. 

In the YF01 phage, the extended MCP isoform is substantially larger (892 a.a) than those seen in many other group members (~500–550 a.a) and most similar in size to that seen in the Paul phage (894 a.a.) [56]. As observed in T3 phage [67], the extended MCP isoform results in the addition of an immunoglobulin (Ig)-like domain to the C-terminus of the MCP. Ig-like domains are extremely common, diverse (>68 domain types recognized), and widely distributed in phages [69]. The specific function of Ig-like domains remains yet to be fully understood; however, they appear to be only present in structural proteins [69], with several pieces of evidence indicating possible involvement in attachment to the host cell surface [69]. The specific implications of these Ig-like domains in *Kuravirus*-like group phages remains unclear. 

Interestingly, the ultrastructural analysis of the group member, SU10 phage, revealed a capsid structure with apparent uniform MCP formation (i.e., lacking the extended isoform) [70]. Consistent with this fact, a band correlating to an extended MCP isoform was absent via SDS-PAGE [70]. This contrasts with earlier work on SU10 phage, which indicated the presence of an extended MCP isoform by mass spectrometry [20]. Upon our inspection of the SU10 phage genome, we noted that while a heptanucleotide slippery sequence is present in the gene encoding MCP, a stop codon (TAG) occurs 39 nucleotides downstream in the −1 frame, leading to the premature termination of the extended MCP isoform, which would result in a product indistinguishable in size (~7 a.a larger) to the natural MCP. This apparent loss of the extended MCP isoform is unique to SU10 phage among the group.

In the middle of this module lie two host lysis-related proteins, a holin (*gp12*) and an endolysin (*gp*13), which work in concert to induce host cell lysis at the conclusion of the phage lifecycle (Figure 6). Many members of the *Kuravirus*-like group have demonstrated good bacteriolytic activity against pathogenic and drug-resistant *E. coli* [17,50,52,53,62], the ability to eliminate *E. coli* from food matrices when used in phage cocktails [50], and have shown effectiveness in reducing *E. coli* load in vivo [53].

The next structural genes in this module encode numerous tail proteins (*gp10-11*, *gp14-16*; Figure 6). The structure and assembly of the SU10 phage provides us detailed insights into how these components may assemble in the YF01 phage, given the high sequence similarity between the two phages [70]. In the YF01 phage, the proximal (*gp10*) and distal (*gp11*) tail fibers likely assemble into hexameric long tail fibers, which are attached to the adapter complex [70]. The nozzle, composed of a hexamer of nozzle proteins (*gp15*), also attached to the adapter complex, is bound by short tail fibers (*gp14*) [70]. The three tail fiber genes showed some of the largest sequence variance across the group (Figure 6), a common finding among related phages [22], and may cause variances in the host range regarding members of the group. Finally, the tail needle, formed by *gp16*, extends from the channel formed by the nozzle.

The final genes in this module appear to encode ejection proteins (*gp18-22*) and may be packaged inside the phage capsid along with the genomic DNA and play roles during the infection process (Figure 6) [70].

#### 3.3.3. DNA Replication and Nucleotide Metabolism

The genome of the YF01 phage contains an assortment of genes involved in DNA replication, repair, and nucleotide metabolism. These include a DNA polymerase (*gp49*) and separately encoded 5′-3′ exonuclease (*gp27*) and 3′-5′ exonuclease (*gp70*) subunits. The separation of the DNA polymerase and its exonuclease domains is common to all group members and is similarly observed in phages closely related to the group (as shown in Figure 4), such as the *Salmonella* phage 7-11, the *Proteus* phage Privateer, the *Vibrio* phage Vp_R1, and the *Aeromonas* phage BUCT695. The YF01 phage also encodes a dual-purpose DNA primase/helicase (*gp71*) and NAD-dependent DNA ligase (*gp57*). The YF01 phage, like many phages, encodes certain enzymes involved in nucleotide metabolism and modification. These include a deoxynucleoside monophosphate kinase (*gp28*), a thymidylate synthase (*gp*60), and a deoxycytidine triphosphate deaminase (*gp68*).

Within the YF01 phage genome, we also identified genes encoding an RNA polymerase sigma factor (*gp30*) and a small transcriptional regular (*gp76*), which are highly conserved across the phage (95.2% and 93.5% average similarity, respectively; Appendix A). Transcriptional studies of the group member phage, phiEco32, highlighted the temporal expression of genes, corresponding to “early”, “middle”, and “late” genes [71]. The expression profiles correspond to the location and function of genes in the phage genome, with early genes located on the far right of the phage genome (*gp82-122* in the YF01 phage; most hypothetical), middle genes located in the middle of the phage genome corresponding to DNA replication genes (*gp23-81* in the YF01 phage), and late genes located at the left of the phage genome corresponding to the virion morphogenesis and lysis genes (*gp1-22* in the YF01 phage). In the phiEco32 phage, RNA polymerase holoenzymes assembled with the phage-encoded sigma factor (*gp30* in the YF01 phage) were shown to drive expression of some middle genes and all late genes (i.e., drive expression later in the phage infection cycle) [71]. The transcriptional regulator (*gp76* in the YF01 phage) appeared to have a dual function in the phiEco32 phage by (1) promoting the expression of the phage sigma factor and (2) shutting off early gene expression via physical interaction with host σ70-factor, essentially promoting the progression of the phage lifecycle to late stage genes [71].

The YF01 phage, like many of the other group phages [17] also encodes a single tRNA (Figure 5; magenta). Phages are known to encode specific tRNA genes to compensate for differences in their codon usage compared to their hosts [72,73]. In the YF01 phage, the tRNA gene has a TCT anticodon corresponding to an arginine (Arg) AGA codon. We analyzed codon usage across all ORFs in the YF01 phage to show AGA was the second most used Arg codon with a usage share of approximately 20.2%. Conversely, in the host *E. coli* strain (K12), AGA was the second least used of five possible Arg codons, with a usage share of approximately 3.6%. This represents a ~5.6 fold increase in AGA codon usage in the YF01 phage over the host (Figure 7; yellow; Appendix A). AGA codon usage shares ranged from ~4.4–5.4 fold across other representative group members from G1-G4 versus *E. coli* K12 Appendix A. While AGA showed the highest fold-increase usage in the YF01 phage, other codons were also found to be preferred when compared to the host (Figure 7; green). It remains unclear why the YF01 phage, *Kuravirus*-like group members, and other phages [74] carry only specific supplemental tRNA genes within their genomes despite the high usage frequency of additional codons. 

### 3.4. The Core Proteome of Kuravirus-like Group Phages

Finally, we wanted to determine the core protein-encoding genome of the *Kuravirus*-like phages. Our pangenome analysis of 32 group members (four phages were excluded due to the presence of minor assembly errors, as assessed by the completeness of the terminase large subunit ORF) revealed a core proteome of 63 proteins, which were shared amongst all member phages (53% of the total proteome in the case of YF01 phage; Figure 8; Appendix A). This core proteome consisted of 16 early genes (all hypothetical or unknown function), 28 middle genes encoding some DNA replication and transcription proteins, and 19 late genes encoding mostly virion structural components (Figure 8). Consistent with the above findings of larger sequence variance across the proximal and distal long tail fibers (and the lack of a clear homologue in some cases) and the short tail fiber, these components were not considered to be core among the group of phages. Conversely, the highly conserved RNA polymerase sigma factor and small transcriptional regulator unsurprisingly formed part of the core proteome. The large terminase subunit, portal protein, MCP (the non-extended isoform), DNA primase/helicase, and a hypothetical protein located in the early genes (no sequence or HMM profile similarity to known domains) were found to show the highest sequence conservation across the group (94.4–96.5% average a.a identity; Appendix A). The phylogenetic reconstruction of the group phages using these proteins as markers resulted in groupings (G1–G4) that were relatively consistent with our earlier whole genome nucleotide (VIRIDIC and VICTOR) or whole proteome (vContact2) analyses, indicating the strength of these five core proteins in phylogenetic analyses Appendix A. 

## 4. Conclusions

This study describes the expansion and reshaping of the members from the current ICTV-classified *Kuravirus* genus from six to thirty-six member phages using a combination of modern bioinformatics approaches—here grouped into four genera, namely *Kuravirus* and the three newly proposed genera, *Vellorevirus*, *Jinjuvirus*, and *Yesanvirus*. We also described a new member of this group, the *Escherichia* phage YF01, isolated from activated sludge in Yokohama, Japan. Genomic analyses described the organization of YF01 phage and characterized the function of encoded proteins involved in phage DNA replication and metabolism, virion morphogenesis, host cell lysis, and transcriptional regulation. Lastly, we determined that sixty-three proteins form the core proteome of *Kuravirus*-like group phages, with five proteins, including the large terminase subunit and DNA primase/helicase, as some of the strongest phylogenetic markers of *Kuravirus*-like phages.

## Figures and Tables

**Figure 1 viruses-15-00506-f001:**
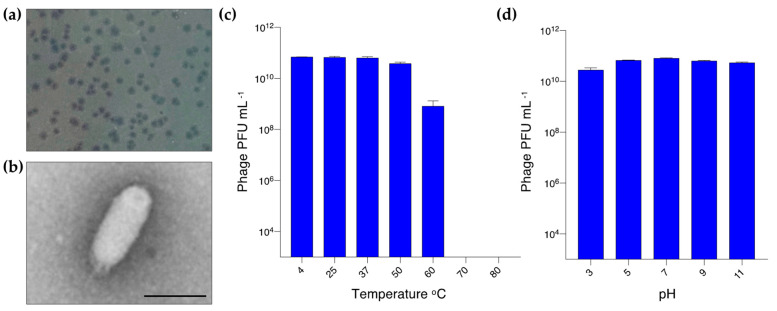
Isolation and properties of the *Escherichia* phage YF01. (**a**) Plaques (~1 mm) produced by the YF01 phage tested on *E. coli* K12 on solid medium. (**b**) Transmission electron microscopy of the YF01 phage. Scale bar, 100 nm. (**c**) Temperature and (**d**) pH stability of the YF01 phage.

**Figure 2 viruses-15-00506-f002:**
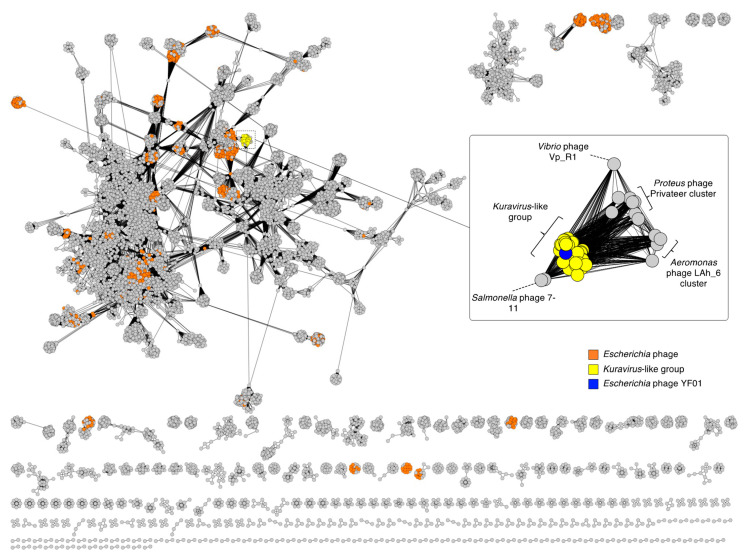
*Kuravirus*-like group phages form a tight cluster using a reticulate network. A reticulate network of >19,000 phage genomes was generated with individual phages represented as nodes (circles) and phage intergenomic similarity represented as edges (lines). *Escherichia* phages are colored in orange. The zoom panel shows the tight clustering of the *Kuravirus*-like group phages (yellow) using an edge-weight spring embedded layout model.

**Figure 3 viruses-15-00506-f003:**
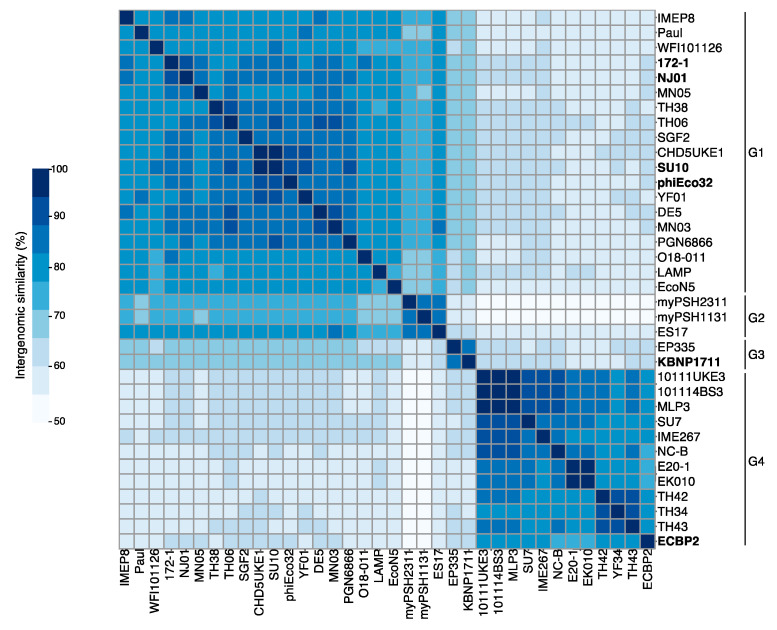
Whole genome nucleotide similarity of phages within the *Kuravirus*-like group. The heatmap demonstrates the splitting of the *Kuravirus*-like group into four genera (shown on right) based on ≥70% intergenomic similarity to cluster into a genus. Bold entries indicate ICTV-classified *Kuravirus* phages. Raw percentage similarity data are shown in the Appendix A.

**Figure 4 viruses-15-00506-f004:**
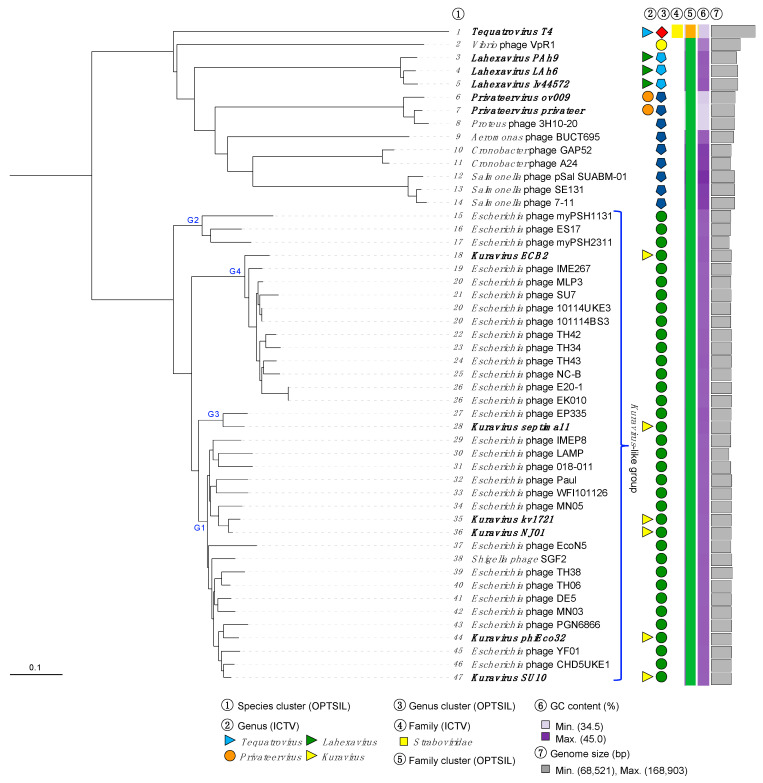
Phylogenetic placement of the *Kuravirus*-like group. Whole genome-based phylogeny (nucleotide) was inferred using Genome-BLAST Distance Phylogeny (100 bootstraps) using the formula D0 yielding an average support of 23%. OPTSIL clustering yielded forty-seven species clusters, five genus clusters and two family clusters. Phages classified by the ICTV are noted in bold. Proposed *Kuravirus*-like group is shown. Scale bar represents the number of nucleotide substitutions per site.

**Figure 5 viruses-15-00506-f005:**
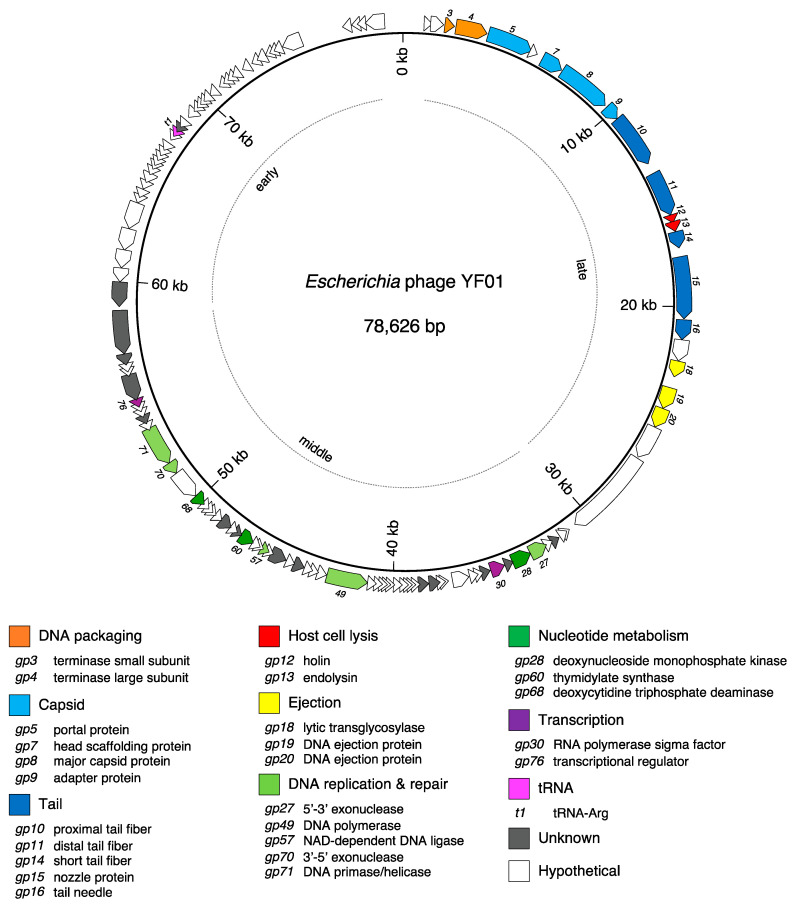
The genome of the *Escherichia* phage YF01. A modular arrangement of genes is noted with those involved in early, middle, and late expression marked. Annotated genes with known function are listed below the map, including genes involved in DNA packaging, phage morphogenesis, host lysis, DNA replication, and metabolism and transcriptional regulation. tRNAs are also shown. Unknown function indicates genes that contain a conserved domain but have no assignable function in the phage lifecycle. Hypothetical indicates genes that do not contain any conserved domains. All YF01 phage genome annotations are shown in the Appendix A.

**Figure 6 viruses-15-00506-f006:**
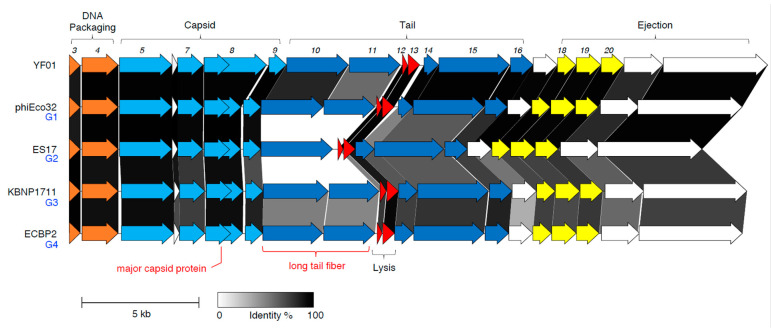
The virion morphogenesis and lysis module of the YF01 phage. Module alignment between YF01 phage and representatives from G1-G4 *Kuravirus*-like group phages demonstrate high gene synteny and identity within the module. The major capsid protein (*gp8*) is expressed in two isoforms (natural and extended) via a ribosomal slippage event. Genes encoding long tail fiber (*gp10-11*), short tail fiber (*gp14*), and nozzle protein (*gp15*) share less sequence identity. Gene color scheme is consistent with that described in Figure 5. Text in red points out relevant differences discussed in the main text. Scale bar represents 5 kb.

**Figure 7 viruses-15-00506-f007:**
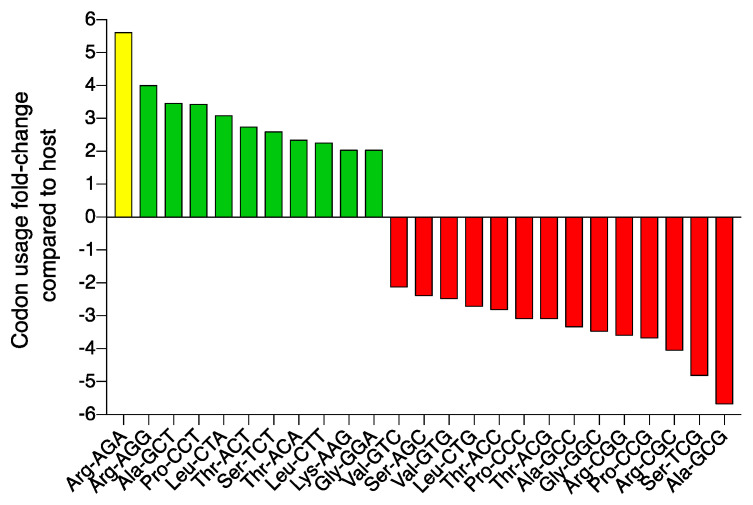
Codon usage bias in the YF01 phage. Codon usage was examined in the coding sequences of both the YF01 phage and *E. coli* K12. Codons whose frequency by two-fold or above were plotted, with positive values (in green) representing codons with higher fold-usage and negative values (in red) representing codons with lower fold-usage in the YF01 phage. The YF01 phage encodes a single tRNA (tRNA^Arg^) with an anti-codon TCT consistent with the higher usage of the Arg-AGA (yellow) codon in the YF01 genome.

**Figure 8 viruses-15-00506-f008:**
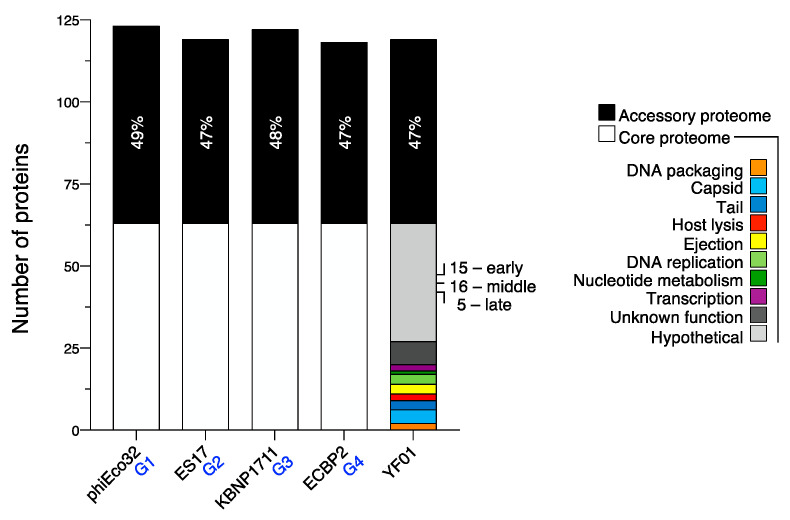
The core proteome of the *Kuravirus*-like group phages. Pangenome analysis indicated 63 proteins are shared by all group phages. Representatives from G1–G4 *Kuravirus*-like group phages are shown alongside the YF01 phage. The core and accessory proteomes of each phage are shown in white and black, respectively, with percentage values indicating the percentage of accessory proteins in each phage. The core proteome of the YF01 phage is further broken down by protein annotation.

**Table 1 viruses-15-00506-t001:** Characteristics of *Kuravirus*-like group phages.

Name	Host	Size (nt)	GC %	CDS ^1^	tRNA ^2^	Geography	Source	Year	Accession	Reference
YF01	*E. coli*	78,626	42.1	121	1	Japan	Wastewater	2021	OQ025076	This study
phiEco32	*E. coli*	77,554	42.3	124	1	USA	Water	2004	EU330206	[17]
ECBP2	*E. coli*	77,315	42.4	119	1	South Korea	Unknown	2012 ^3^	JX415536	[18]
NJ01	*E. coli*	77,448	42	132	1	China	Animal	2012 ^3^	JX867715	[19]
KBNP1711	*E. coli*	76,184	42.4	124	1	South Korea	Unknown	2013 ^3^	KF981730	
SU10	*E. coli*	77,327	42.1	124	1	Sweden	Unknown	2014 ^3^	KM044272	[20]
172-1	*E. coli*	77,266	42	128	1	China	Animal feces	2014 ^3^	KP308307	
EK010	*E. coli*	78,078	42.1	120	1	China	Wastewater	2020 ^3^	LC553734	[50]
O18-011	*E. coli*	75,646	42.1	122	1	China	Unknown	2020 ^3^	LC553735	[50]
LAMP	*E. coli*	68,521	42.2	96	1	Russia	Animal feces	2017 ^3^	MG673519	
EP335	*E. coli*	76,622	42.5	123	1	Netherlands	Wastewater	2018 ^3^	MG748548	[51]
myPSH2311	*E. coli*	68,712	42.3	118	0	India	Wastewater	2018 ^3^	MG976803	[52]
myPSH1131	*E. coli*	76,163	42.4	129	0	India	Water	2018 ^3^	MG983840	[53]
NC-B	*E. coli*	76,641	42.1	116	1	USA	Human feces	2019 ^3^	MK310183	[54]
WFI101126	*E. coli*	77,307	42.1	130	1	Germany	Sewage	2015	MK373770	[55]
Paul	*E. coli*	79,429	42	124	1	USA	Water	2018	MN045231	[56]
SGF2	*S. flexneri*	76,964	42.3	118	1	China	Water	2019 ^3^	MN148435	[57]
ES17	*E. coli*	75,007	42.1	120	1	USA	Sewage	2018	MN508615	[58]
EcoN5	*E. coli*	76,083	42.1	128	1	Colombia	Unknown	2019 ^3^	MN715356	
PGN6866	*E. coli*	78,549	42.3	129	1	India	Sewage	2020 ^3^	MT127620	
MN03	*E. coli*	77,187	42.2	124	1	Bangladesh	Water	2017	MT129653	
MN05	*E. coli*	76,899	42.2	126	1	Bangladesh	Water	2017	MT129655	
TH06	*E. coli*	77,678	42.1	123	1	USA	Wastewater	2020 ^3^	MT446386	[59]
TH34	*E. coli*	77,944	42.3	120	1	USA	Wastewater	2020 ^3^	MT446407	[59]
TH38	*E. coli*	81,552	42.2	132	1	USA	Wastewater	2020 ^3^	MT446410	[59]
TH42	*E. coli*	77,284	42.3	118	0	USA	Wastewater	2020 ^3^	MT446413	[59]
TH43	*E. coli*	77,980	42.4	118	1	USA	Wastewater	2020 ^3^	MT446414	[59]
DE5	*E. coli*	77,305	42.1	125	1	China	Unknown	2021 ^3^	MW741821	
101114BS3	*E. coli*	75,747	42	113	1	Austria	Wastewater	2018	MZ234015	[60]
101114UKE3	*E. coli*	75,747	42	113	1	Austria	Wastewater	2018	MZ234017	[60]
CHD5UKE1	*E. coli*	77,359	42.2	123	1	Austria	Wastewater	2018	MZ234028	[60]
SU7	*E. coli*	76,626	42.1	117	1	Sweden	Wastewater	2016	MZ342906	[61]
IME267	*E. coli*	76,631	42	117	1	China	Unknown	2021 ^3^	MZ398243	
IMEP8	*E. coli*	75,809	42.1	118	1	China	Animal milk	2021	MZ648214	
MLP3	*E. coli*	76,234	42.1	115	1	Chile	Water	2019	OK148440	[62]
E20-1	*E. coli*	77,938	42.2	122	1	China	Wastewater	2018	OP293233	

^1,2^ CDS and tRNA numbers are reported from the Prokka annotation pipeline for consistency. ^3^ Unknown year of isolation, Genbank submission date listed.

## Data Availability

The genome sequence of the *Escherichia* phage vB_EcoP-YF01 is available in NCBI Genbank under the accession number OQ025076. Raw sequence data are available under BioProject accession number PRJNA912594.

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
