# Peer review of "Expansion of Kuravirus-like Phage Sequences within the Past Decade, including Escherichia Phage YF01 from Japan, Prompt the Creation of Three New Genera"

_viruses, 2023, doi:10.3390/v15020506_

Round 1

Reviewer 1 Report

The manuscript describes a new E. coli virus and suggests reshaping the Kuravirus genus.  The manuscript is very well prepared; data are very precisely analyzed using different bioinformatic tools. I have just a small remark: check again the writing of Kuravirus; sometimes it is in italic, and sometimes not. This is a Latin name that principally should be in italic. The supplementary files could not be opened. 

Reviewer 2 Report

This manuscript provides a genome-wide characterization of a bacteriophage isolated from a sewage treatment plant. This manuscript has a logical order and presents an affluent description of results. However, I consider that there are several methodological aspects to be improved and/or described so that the manuscript is consistent with the stated objectives.

1) The title does not faithfully reflect what is delivered as a result. Even more so when the objective of the manuscript is somewhat confusing and requires improvement in its wording.

2) It is strange that only one isolation of bacteriophages from environmental samples is mentioned. I suggest providing a clear argument as to why the phage named YF01 was selected.

3) The authors use a single isolate of E. coli (K-12) to demonstrate the specificity of phage YF01, without concrete evidence of the actual host range. Although there are changes in the nomenclature used by ICTV, this information is key to its preliminary classification.

4) Much of the experimental work is based on bioinformatic analysis (genomic and later proteomic) from MinION sequencing of phage YF01. However, it has not yet been a validated strategy for sequencing bacteriophage genomes. At present, parallel sequencing via Illumina is performed to corroborate the sequencing and to be able to faithfully perform comparative analyses against public databases. For this reason, it is strongly suggested to include this information in the respective analyses, especially for regrouping.

5) I consider that obtaining a direct access number to the genome of YF01 in the NCBI database is essential to analyze the relevance of generating subgroups within the genus Kuravirus. 

Reviewer 3 Report

The current work is an interesting study which reports the characterization of a novel Escherichia phage YF01 (isolated from activated sludge of a wastewater treatment plant in Yokohama, Japan). The authors have sequenced the phage genome by Nanopore long read technology and shown that the phage is related to the ICTV-classified Kuravirus genus. They characterized the pH and temperature stability of the isolated phage. They performed phylogenetic analysis by using a network-based approach (vContact2) and VIRIDIC, which led to discrepancy of genera groupings. They used VICTOR for reconstruction of whole genome phylogenetic tree of clustered phages. They also analyzed the genome organization and annotation of the novel phage and performed pangenomic analysis to identify its core proteins. According to the findings, they suggest the expansion and reshaping of Kuravirus genus into 36 member phages (Kuravirus group) and describe a new member of the expanded group.

There are some concerns that need to be addressed in the manuscript. 

Line 32: All phages do not lyse bacteria.

Add a brief explanation to Introduction or Discussion about Nanopore sequencing and its importance in genomic studies of bacterial viruses, mentioning some references which have used this technology for phage genome studies.

Lines 77 and 80: Add the country for Advantec Toyo.

Line 82: Mention how many plaques were removed?

Line 89-90: Add how many dilutions were included for plaque count and titer calculation?

Lines 90-91: Which pH was used for temperature testing?   

What was the starting titer for temperature and pH stability studies? Clarify it in Materials and Methods.

In Figure 1; The morphology of phage YF01 is not clear. This picture is suggested to be replaced with a picture of higher quality in which the morphology (morphological characteristics) of phages can be detected clearly. 

Some phages do not have any reference in Table 1. Add references for them.

There is no Chapter 3.2 in Results.

Explain the advantage of vContact2 for taxonomy assignment.

Line 203: It is not clear which criteria, tools, etc were used to extract phages that share a relationship with YF01 phage. Clarify in the text.

Line 237: Why did you use VICTOR for reconstruction of phylogenetic tree? What is the advantage of VICTOR over other packages and tools? Briefly explain in the text.

Make it more clear and discuss the reason(s) for contradiction observed between OPSTIL and vContact2/VIRIDIC analyses. Why is this contradiction observed?

Line 355: It is not correct to use the reference 67, which is about SU10 phage, for YF01 phage. This reference is already cited above.

Line 398: How was the codon usage analysis was performed? Which tool, database, ect? It should be mentioned in the manuscript.

Add numbering (2.1, 2.2, etc) to the sub-sections of Chapter 2 (Materials and methods)

There are some typos in the manuscript, lines 116, 171, 334, 380, among others.

Reviewer 4 Report

The manuscript entitled: “Expansion of the Enterobacteriaceae Kuravirus phage group to 36 members”, reference: viruses-2143314

The manuscript is highly interesting, well written thus, easy to follow, and displays a high scientific soundness. The presented proposition is well supported.

Nevertheless, there are some key points that would like to address:

Line 314 to 317, in my opinion this statement should be corroborated by a transmission electron microscopy image of the bacteriophage. In my opinion this is a serious Achilles heel of this manuscript. Is it possible the inclusion of its image? It is, in my point of view, mandatory for an accurate and robust classification of the bacteriophage. Do the authors understand my point of view?

Moreover, the authors clearly state that it is a Escherichia coli bacteriophage, nevertheless, no host range was tested correct? So we are not certain of its lytic spectrum, correct?

 Finally, as a request, can the authors please provide the information on what is the main type of effluent received by the wastewater treatment plant? Is it urban? Industrial? Rural?

Point by point comments:

Line 70, please consider capitalize the unit for liter as: g L-1, just like as used in line 78. Please thoroughly revise the entire manuscript.

Line 76, the unit for gravitational force is an italicized “g”. It does not have the “x”.

Line 90, please separate the temperature unit from its numerical value. Please do so for all units except percentage.

Line 102, the percentage of ethanol is (v/v)? Please revise all similar units.

Reviewer 5 Report

The manuscript entitled "Expansion of the Enterobacteriaceae Kuravirus phage group to 36 members" highlights the isolation and characterization of Escherichia phage isolated from wastewater in Japan. 

Furthermore, the authors' analysis demonstrated the identification of these phages as Kuravirus phages, and the authors suggested the expansion of Kuravirus phages to 36 members. 

The article is well written with appropriate methodology and well designs analysis. The study also shows the novel isolation of new phages from wastewater. 

I suggest authors discuss a little bit about the use of these phages particularly in terms of combating developing antimicrobial resistance specifically in E. coli related to food and waterborne illness caused by pathogenic strains of E. coli

Round 2

Reviewer 4 Report

I must congratulate the authors for improving the manuscript quality. The TEM image is enough for me. The long tail is not clearly perceivable, but, in my opinion, there is an indication of its presence. Hopefully, in a future publication the authors include a clearer TEM figure. The range of hosts is in my opinion highly relevant for potential future applications, and to encompass a more robust identification of the bacteriophage. Nevertheless, I agree with of the authors, it does not in any way impede relevance of this publication.